# Standardization of Semi-Quantitative Dot Blotting Assay—Application in the Diagnosis, Follow-Up, and Relapse of Paracoccidioidomycosis

**DOI:** 10.3390/microorganisms12020351

**Published:** 2024-02-08

**Authors:** Beatriz Aparecida Soares Pereira, Ricardo de Souza Cavalcante, Vera Lucia Pereira-Chioccola, Marcia de Souza Carvalho Melhem, Lídia Raquel de Carvalho, Rinaldo Poncio Mendes

**Affiliations:** 1UNESP Botucatu, School of Medicine—Discipline of Infectology, São Paulo State University, Botucatu 18618-689, São Paulo State, Brazil; ricardo.cavalcante@unesp.br (R.d.S.C.); melhemmr@uol.com.br (M.d.S.C.M.); tietemendes@terra.com.br (R.P.M.); 2Adolfo Lutz Institute, São Paulo 01246-000, São Paulo State, Brazil; pchioccola@gmail.com; 3FUMS Campo Grande, School of Medicine, Federal University of Mato Grosso do Sul, Campo Grande 79070-900, Mato Grosso do Sul, Brazil; 4Institute of Biosciences—Department of Biodiversity and Biostatistics, UNESP Botucatu, São Paulo State University, Botucatu 18618-689, São Paulo State, Brazil; lidia.carvalho@unesp.br

**Keywords:** paracoccidioidomycosis, dot blotting, diagnosis, serology, relapse

## Abstract

Introduction: This study standardized a semi-quantitative dot blotting assay (DB) and a quantitative real-time polymerase chain reaction (qPCR) to detect specific antibodies for *Paracoccidioides brasiliensis* and its DNA in PCM patients. Methodology: We evaluated 42 confirmed PCM patients upon admission using a serological double agar gel immunodiffusion test (DID), DB, and molecular tests (qPCR in total blood). The control groups included 42 healthy individuals and 37 patients with other infectious diseases. The serological progress during treatment was evaluated in eight patients, and there was a relapse diagnosis in ten patients using the Pb B.339 strain antigen. The cut-off points for the serological tests were determined by a receiver operator characteristic curve. Results: The DB and DID tests showed similar accuracy, but the DB identified lower antibody concentrations. Cross-reactions were absent in the DB assay. In the relapse diagnoses, DB exhibited much higher sensitivity (90%) than DID (30%). Conclusions: A DB assay is easier and faster than a DID test to be performed; DB and DID tests show the same accuracy, while blood qPCR is not recommended in the diagnosis at the time of admission; cross-reactions were not observed with other systemic diseases; DB and DID tests are useful for treatment monitoring PCM patients; and a DB assay is the choice for diagnosing relapse. These findings support the introduction of semi-quantitative DB assays in clinical laboratories.

## 1. Introduction

Paracoccidioidomycosis (PCM) is an endemic and systemic mycosis caused by fungi of the *Paracoccidioides brasiliensis* complex and by *Paracoccidioides lutzii* [1,2], which live saprophytically in the soil of Latin America, mainly Brazil, Argentina, Colombia, and Venezuela [2]. This disease afflicts individuals with present or previous intense contact with soil, such as rural workers [3].

Active PCM has two main clinical forms—the acute/subacute form (AF) and the chronic form (CF). The AF, also called the juvenile form, affects children, adolescents, and young adults and is responsible for 15% to 20% of cases, with a male/female ratio of 1.7:1.0. The AF has a low duration of symptomatology—usually less than two months—and is characterized by the involvement of organs rich in the phagocytic mononuclear system—lymph nodes, the spleen, the liver, and bone marrow; pulmonary lesions are rare (<5%), and the mucous membranes of the upper airway and digestive tract (UADT) are affected in less than 20% of cases. Regarding its severity, this clinical form is classified as moderate or severe [2,4]. The CF, also called the adult form, afflicts adults usually over 30 and has a duration of symptomatology higher than four months and frequently higher than six months. It is responsible for 80% to 85% of cases and has a male-to-female ratio of 22.0:1.0. The CF affects the lungs in almost all cases, and the mucous membranes of the UADT are frequently affected. When present, lymph node involvement is observed in the cervical and submandibular chains, usually not so intense. Skin lesions have been observed in both the AF and CF, predominantly in the face [2,4].

The gold standard for PCM diagnosis is the recovery and identification of the etiologic agent in clinical specimens using mycological, cytopathological, and histopathological examinations [5]. However, the time necessary to achieve the identification results by culture and the invasiveness of the biopsy to collect tissue for histopathological examination, associated with the possibility of identifying specific serum antibodies, stimulate the use of serological tests for diagnosing the disease. In the III International Meeting on Paracoccidioidomycosis, held in Medellín (Colombia) in 1986, the double agar gel immunodiffusion test was chosen as the reference method to evaluate antibody serum levels, using an antigen to be standardized by Dr. Zoilo Pires de Camargo [5]. In addition, serological tests have been useful in evaluating patient severity [6] and in the follow-up of patients under antifungal treatment as a fundamental criterion of cure [2].

DID has been used in the routine diagnosis of PCM because it is easy to perform, has a good cost–benefit relationship, and shows high specificity, positive predictive values, and positive and negative likelihood ratios. However, some limitations have been observed, including the identification of antibodies in the cerebrospinal fluid in the serum of immunosuppressed patients, such as those with AIDS [6,7,8], cancer, and several inflammatory diseases [9], and cases of relapse in non-immunosuppressed PCM patients [10]. These limitations have stimulated an assessment of the semi-quantitative dot blotting (or dot blot—DB) assay and the real-time polymerase chain reaction (qPCR) in the diagnosis and in the follow-up of PCM patients. The DB assay, used to identify monoclonal antibodies, soluble proteins, nucleic acids, and some infectious diseases, was evaluated a few times in PCM patients. Likewise, qPCR was evaluated a few times in PCM diagnosis [11,12,13,14,15].

### 1.1. Patients and Methods

Study setting: This study was conducted at the University Hospital of the São Paulo State University, UNESP Botucatu, School of Medicine (UBSM), Botucatu (São Paulo state, Brazil). PCM patients were admitted to the Tropical Diseases Ward “Domingos Alves Meira” and/or assisted by the Medical Mycology Outpatients Service from the University Hospital. Mycological and serological procedures were conducted at the Laboratory of Infectious and Parasitic Diseases—Medical Mycology within the experimental research unit (Unipex) at the same institution.

### 1.2. Patients and Controls

This study was conducted with 60 PCM patients; 42 were treatment-näive, 8 were under antifungal treatment follow-up (four with AF and four with CF), and 10 were undergoing a relapse. All the patients had PCM confirmed by the mycological and/or histopathological identification of the etiologic agent (confirmed cases) or diagnosed through the identification of specific serum antibodies determined by the DID test (probable cases), according to the specifications of Mendes et al. (2017) [2]. The patients were classified into clinical forms as AF or CF; they were also classified according to severity—AF as moderate or severe and CF as mild, moderate, or severe, also following Mendes et al. (2017) [2]. A case was defined as a relapse when, after efficacious treatment characterized by reaching all the criteria of cure, a patient presented new signs and symptoms associated with identifying the typical *Paracoccidioides* yeast cells in clinical specimens. The control group consisted of 42 blood donors from the Blood Bank of Botucatu, located at UBSM, living in the same region as the patients. Additionally, ten patients with classic histoplasmosis, ten with tuberculosis, ten with aspergillosis, and seven with cryptococcosis, diagnosed by the microbiological and/or serological identification of the etiological agent in clinical materials, were assessed. The serum samples collected from these patients with active disease at the time of admission were subjected to a cross-reactivity evaluation.

Exclusion criteria: Patients with other infectious, inflammatory, or neoplastic diseases as comorbidities, as well as pregnant and lactating females, were excluded from the study. Smoking and alcohol consumption were not considered exclusion criteria.

Patient follow-up: The initial or induction treatment was maintained until a clinical cure was observed. A clinical cure was characterized by the disappearance of signs and symptoms presented at the patient’s admission, except those related to possible sequelae, and by the normalization of the erythrocyte sedimentation rate (ESR). The normalization of serum γ-globulin levels, determined by serum protein electrophoresis, was the reference variable used for patients with no increased ESR at the time of admission. During the initial treatment, the patients were subjected monthly to clinical, radiological, parasitological, urinary, serological, and blood biochemistry evaluations. The complementary treatment phase involved clinical, serological, and radiological evaluations every three months, lasting for up to one year after consistently negative serological test results were observed [2].

## 2. Methods

### Study Design

The present study consisted of six independent experiments, as shown in the infographic presentation (Figure 1).

## 3. Blood Collection and Serum Sample Storage

Blood samples were drawn by venous puncture using tubes with a coagulation activator, silica, and a gel separator (5 mL Vacuplast) to obtain serum. Whole blood samples were collected in tubes with EDTA (5 mL Vacuplast). All samples were divided into aliquots, stored in an ultra-freezer at −80 °C, and kept in the Unipex biobank under careful temperature control.

### 3.1. Serum Samples from PCM Patients and Controls

Circulating anti-*P. brasiliensis* antibodies were investigated in 182 serum samples, distributed as follows: (a) 42 samples from treatment-naïve PCM patients; (b) 42 samples from healthy blood donors; (c) 38 samples from four AF patients undergoing antifungal treatment follow-up; (d) 40 samples from four CF patients undergoing antifungal treatment follow-up; and (e) 20 paired samples from 10 patients who experienced relapse, which were drawn at the time of admission—pre-treatment and at the time of relapse.

### 3.2. Serum Samples from Patients with Other Active Infectious Diseases for Cross-Reactivity Evaluation

Serum samples from 37 patients were evaluated, distributed as follows: 10 from histoplasmosis patients, confirmed by histopathological examination; 10 from patients with chronic pulmonary aspergillosis, confirmed by histopathological examination and/or DID test; 10 from pulmonary tuberculosis patients, confirmed by identification of acid-fast bacilli in the sputum, using Ziehl–Neelsen staining; and 7 from cryptococcosis patients, confirmed by the identification of the etiologic agent by India ink staining and/or the latex agglutination test.

## 4. Antigen Preparation—*P. brasiliensis* Yeast Phase Filtrate

The antigen used in the serological tests was a yeast phase filtrate from *P. brasiliensis* B.339 (Pb B.339), prepared at the Laboratory of Infectious and Parasitic Diseases—Medical Mycology (Unipex), according to the specifications of Camargo et al. (2003) [5]. The protein content of the antigen was determined using a Nanodrop instrument (NanoVue Measuring Nucleic Acids and Proteins—GE Healthcare Life Sciences AB, Björkgatan, Sweden). Three measurements of the antigen preparation were taken, and the protein concentration was determined by calculating the average of the values obtained. This antigen was used in the DID test and also in the semi-quantitative DB assay.

## 5. Double Agar Gel Immunodiffusion Test

The DID test was performed following the specifications of Restrepo [16] and stained with 10B starch black (Dinâmica). The serum was considered a reagent when the formation of precipitation line(s) was observed, and the serum level was defined by the highest dilution at which these lines were identified.

## 6. Standardization of the Semi-Quantitative Dot Blotting Assay

The semi-quantitative DB assay was standardized based on procedures designed by Kamikawa et al. (2017) [14] and Pappas et al. (1983) [17]. Briefly, in this process, 0.22 μm nitrocellulose membranes (Bio-Rad, Hercules, CA, USA) were sensitized with the antigen obtained from the yeast phase of the Pb B.339 strain. Then, the membranes were exposed to the serum samples from the PCM patients, and a secondary antibody was used—a goat anti-human IgG antibody conjugated with peroxidase (Sigma, Kawasaki, Japan) at a concentration of 1:2000. The reaction was developed using the DAB chromogen kit (Scytek, Logan, UT, USA).

## 7. Quantitative Real-Time Polymerase Chain Reaction

The qPCR for the detection of *P. brasiliensis* DNA was performed according to the specifications of Buitrago et al. (2009) [15], under the guidance of Dr. Vera Lúcia Pereira-Chioccola. The reactions were conducted on 49 DNA samples, and 7 of them were from treatment-naive patients. The primers and probe amplified an ITS1 region of the ribosomic DNA. The designer included the forward (OliPbMB1) 5′-ACCCTTGTCTATTCTACC-3′ and the reverse (OliPbMB2) 5′-TTACTGATTATGATAGGTCTC-3′ primers and the PbMB1 probe 5′-CGCGATCGCCGGGGACACCGTTGAAATCGCG -3′ labeled with FAM and BHQ1 at the 5′ and 3′ ends, respectively. The reactions were performed in an AreaMx Real-Time PCR System (Agilent, Santa Clara, CA, USA) with a final volume of 20 µL per reaction. In each tested sample, DNA (3 µL) or control DNA (1 µL) was added to a mixture containing 10 µL of 2× TaqMan Universal PCR Master Mix (Thermo Scientific, Waltham, MA, USA) and the primer set (1.25 µL of an Assay Mix containing 18 µM of each molecular primer and 5 µM of the probe). The qPCR analysis and setting of the parameters were performed using the AreaMax-Real-Time PCR software, version 2.1.

The blood used for the qPCR was collected from the same patient; at the same time, a fraction was used for serum separation to perform the serological evaluations, i.e., the DID test and the semi-quantitative DB assay.

## 8. Cut-Off Point and Accuracy Parameters

The cut-off point for each serological test was determined by the receiver operator characteristic (ROC) curve, and their respective accuracies were calculated following Fletcher and Fletcher [18].

## 9. Data Presentation and Statistical Analysis

As the results of the serological tests were presented as dilutions of the serum samples with a ratio of two, they were converted into scores (Table 1). The following statistical tests were used to analyze the results: the Cochran Q test for comparing more than two dependent variables, the McNemar test for comparing two dependent variables, and a profile analysis for comparing the evolving titers of two tests in relation to the treatments. The null hypothesis was rejected with a type I error equal to or less than 5% (*p* ≤ 0.05). The statistical tests were performed using SPSS version 9.4.

## 10. Results

### 10.1. Experiment 1—Standardization of Semi-Quantitative Dot Blotting Assay

Materials, solutions, and equipment: To perform the semi-quantitative DB assay, we used (a) a 96-well flat-bottom microplate; (b) a 0.22 µm nitrocellulose membrane (Bio-Rad); (c) a 6 mm-diameter paper punch (Paperpro, USA); (d) PBS at a pH of 7.4 (Sigma); (e) an antigen obtained from the Pb B-339 strain; (f) 5% skimmed milk in PBS (Molico, São Paulo, Brazil) as a blocking solution; (g) serum from the PCM patients; (h) 3% skimmed milk in PBS (Molico); (i) a goat anti-human IgG antibody conjugated with peroxidase (Sigma) at a dilution of 1:2000 as a secondary antibody; (j) PBS with 0.1% Tween 20; (k) an incubator oven (Quimis, Brusque, Brazil); (l) an orbital shaker (Fanen, São Paulo, Brazil, model 255-B); (m) a single-channel pipette (Eppendorf, Hamburg, Germany) and a 12-channel multichannel pipette (Eppendorf); (n) aluminum foil; and (o) a DAB chromogen kit (Scytek).

### 10.2. Antigen

Three measurements of the antigen’s protein content were taken: the first was 9.29 ng/µL, the second was 8.09 ng/µL, and the third was 7.95 ng/µL. The average of these three measurements resulted in an antigen protein content of 8.44 ng/µL.

The antigen was tested undiluted in PBS and diluted in PBS at a pH of 7.4 in a 1:2 ratio. A high-titer control serum and a low-titer control serum were used for the semi-quantitative DB assay to visualize the possible shades that positive samples could exhibit, with a chosen dilution of 1:8. To standardize and determine the ideal volume of antigen applied to the membrane, tests were conducted with volumes of 1 µL, 2 µL, 3 µL, and 6 µL. The volume that yielded the best result was 2 µL of antigen per dot at 4.88 ng/µL. The nitrocellulose membranes were sensitized with 2 µL of the filtered culture antigen obtained from the Pb B-339 strain, diluted in PBS at a pH of 7.4 at a ratio of 1:8. Subsequently, the microplate was incubated in an oven at 37 °C for 30 min to adsorb and fix the antigen.

Using the antigen diluted at a ratio of 1:8 showed that the previous cross-reaction observed with the undiluted antigen was not obtained again.

### 10.3. Semi-Quantitative Dot Blotting Protocol

Nitrocellulose membrane discs with a pore size of 0.22 µm (Bio-Rad) were sensitized with 2 µL of the antigen preparation obtained from the Pb B-339 strain using a single-channel pipette (Eppendorf). After sensitization, the microplate was incubated in an incubator oven (Quimis) at 37 °C for 30 min to fix the antigens to the nitrocellulose membrane.

Subsequently, the membranes were blocked by adding 100 µL of 5% skimmed milk in PBS (Molico) to each well using a 12-channel multichannel pipette (Eppendorf). The microplate was constantly agitated using an orbital shaker (Fanen, São Paulo, Brazil, model 255-B) for one hour at room temperature.

The blocking solution was then aspirated using a 12-channel multichannel pipette (Eppendorf). Subsequently, 50 µL of undiluted and diluted sera in 3% skimmed milk in PBS (diluted at a 1:2 ratio) were pipetted, following the working map previously identified with serum numbers and their dilutions. The material was incubated for two hours at room temperature with continuous agitation.

The contents were aspirated using a 12-channel multichannel pipette (Eppendorf), followed by three five-minute washes with 100 µL of 0.1% PBS-Tween 20 per well. Subsequently, 50 µL of the secondary antibody (the goat anti-human IgG antibody conjugated with peroxidase (Sigma)) at a dilution of 1:2000 was pipetted into each well using a single-channel pipette (Eppendorf). The microplate was covered with aluminum foil to prevent exposure to light and incubated for one hour and 30 min with continuous agitation.

The secondary antibody was aspirated, followed by three five-minute washes with 100 µL of 0.1% PBS-Tween 20. After this step, 50 µL of the DAB chromogen was pipetted into each well using a single-channel pipette and left for 10 min. This was followed by a wash with distilled water to remove non-specific color reaction residues. Strong and weak circles with a well-defined brown color were considered reactive, while the absence of color was considered non-reactive.

Figure 2 shows the different phases of the semi-quantitative DB assay.

### 10.4. Experiment 2—Standardization of the Quantitative Real-Time Polymerase Chain Reaction

The qPCR specifications of Buitrago et al. [15] could be reproduced in our laboratory. However, when it was performed in blood samples, the positivity was very low—only four out of forty-two patients (9.5%). Thus, the procedure was also performed in concentrated blood samples from seven patients, but the results were persistently negative.

### 10.5. Experiment 3—Evaluation of the Positivity of the Dot Blotting Assay, the Double Agar Gel Immunodiffusion Test, and the Quantitative Real-Time Polymerase Chain Reaction in the Diagnosis of Paracoccidioidomycosis

The positivity of the serological tests and the molecular diagnosis were evaluated in the 42 PCM patients with an active disease before treatment, characterized by sex, age, clinical form, and severity (Table 2).

Among the affected organs in the AF, lymph nodes were the most prevalent (87.5%), compared to the other organs: lungs (25%), spleen (25%), UADT (12.5%), liver (12.5%), and bones (12.5%) (*p* = 0.0007; Figure 3A). In the CF, pulmonary involvement was the most prevalent (91%), followed by UADT (35%), skin (23%), larynx (18%), lymph nodes (9%), adrenal glands (9%), the central nervous system (CNS) (6%), genitals (6%), and bones (3%) (*p* < 0.0001; Figure 3B).

The ROC curves for both the DID test (Appendix A and Figure 4A) and DB assay (Appendix A and Figure 4B) demonstrated that the cut-off point was the undiluted serum.

The positivity observed was 78.6% for the DB assay, 73.8% for the DID test, and 9.5% for the qPCR; the serological reactions showed higher sensitivity than the molecular test ((DB = DID) > qPCR; *p* < 0.0001).

The comparison of the titers in terms of AF severity showed that the moderate and severe patients had no difference in either the DID test or DB assay. However, for the CF patients, the mild cases showed lower titers than the moderate and severe ones, which, in turn, did not differ from each other ((moderate = severe) > mild) in either the DID test or DB assay (DID, *p* < 0.005; DB, *p* < 0.008) (Table 3).

### 10.6. Experiment 4—Determination of the Accuracy Parameters of the Double Agar Gel Immunodiffusion Test and the Dot Blotting Assay in Paracoccidioidomycosis Patients

The accuracy parameters were calculated for tests performed in 42 patients with active PCM at the time of admission (pre-treatment) and in the sera of 42 healthy individuals (blood donors) from the same region where the patients lived. The results of these tests are shown in Table 4.

The accuracy parameters of the serological DB test and DID assay (Table 5) showed the strong similarity of the results, highlighting their specificity, positive predictive value, and positive and negative likelihood ratios.

Cross-reactions: the sera from the patients with histoplasmosis, cryptococcosis, tuberculosis, and aspergillosis did not show a cross-reaction in the standardized DB assay using antigen from the Pb B.339 stain diluted at a ratio of 1:8.

### 10.7. Experiment 5—Evaluation of Serological Tests: Dot Blotting Assay and Double Agar Gel Immunodiffusion Test in the Follow-Up of Treated Patients

The evaluation of the serological tests in the follow-up of patients undergoing antifungal treatment for PCM was conducted in eight patients, four of them with AF and four with CF, whose distribution according to clinical form, severity, gender, and age is shown in Table 6.

The four AF patients evaluated during the treatment follow-up had lymph node involvement (three cases) as well as involvement in the lungs, skin, liver, spleen, and bones (one case each). The four CF patients, also assessed during the treatment follow-up, showed pulmonary (four cases), UADT (three cases), and trachea (one case) involvement.

The serological titers determined by the DB assay and DID test decreased during the antifungal treatment, which was best fitted by a polynomial curve (see Appendix A for each AF patient, Appendix A for each CF patient, Figure 5A for the four AF patients, Figure 5B for the four CF patients, and Figure 5C for all eight patients).

The titers of the specific antibody serum levels were analyzed, comparing the curves observed with the DB assay and the DID test and the titers in response to the treatment. For the AF patients, the results showed *p* < 0.0001 and *p* < 0.0001, for DB and DID, respectively; for the CF patients, the results showed *p* = 0.002 and *p* < 0.0001, respectively; and for all eight patients (AF + CF), the results showed *p* = 0.001 and *p* < 0.0001, respectively.

Therefore, the DB assay gave higher titers than the DID test for the AF patients; however, the DID serum progressed to a non-reagent in the four patients, while the DB serum progressed to a non-reagent in two cases and to an undiluted reagent in the other two during the follow-up period evaluated (Appendix A). In addition, the CF patients also had higher titers in the DB assay than the DID test, although while the DID serum progressed to a non-reagent in the follow-up period studied, the DB serum persisted as a reagent (Appendix A). Evaluating patients with both clinical forms showed the persistence of the reagent reactions in the DB assay.

### 10.8. Experiment 6—Evaluation of the Serological Tests: Double Agar Gel Immunodiffusion Test and Dot Blotting Assay in the Diagnosis of Relapse

The semi-quantitative DB assay in patients with a relapse was evaluated for 10 patients, whose distribution according to gender, age, clinical form, and severity at the first admission is shown in Table 7.

Pulmonary involvement was the most prevalent (n = 9; Cochran Q test; *p* < 0.0001). The lymph nodes (n = 5), larynx (n = 4), skin (n = 4), UADT (n = 1), intestines (n = 1), and CNS (n = 1) were also affected.

The DID test showed a positivity of 80% at the pre-treatment stage and 30% during the relapse (*p* = 0.07), while the DB assay was positive in 100% of cases at the pre-treatment stage and 90% during the relapse (*p* = 1.00). The comparison of the DID and DB tests revealed that the positivity values did not differ (*p* = 0.48) at the pre-treatment stage; on the other hand, the DB assay was more sensitive than the DID test (*p* = 0.04) during the relapse (Table 8).

## 11. Discussion

The gold standard of PCM diagnosis is identifying the etiological agent in clinical specimens using direct mycological, cytopathological, and histopathological examinations, as well as isolation in culture [2]. Clinical specimens are easily obtained when the affected organs are the lungs by sputum collection or when the mucous membranes of the upper airways, superficial lymph nodes, and skin are affected by fine needle aspiration, tissue scraping, or a biopsy [19]. The identification of typical *Paracoccidioides* spp. yeast forms confirms the diagnosis. However, a biopsy is an invasive procedure contraindicated in some cases and avoided by many patients and even by some physicians. In addition, isolation in culture at room temperature yields results only after some weeks due to the slow growth of this fungus, followed by the confirmation of its thermal dimorphism—a demonstration of the transition to the yeast form when cultivated at 37 °C. These challenges have discouraged the use of the method of isolating the fungus in culture and histopathological examinations in the routine of many clinical laboratories.

At the same time, several serological tests for identifying specific antibodies have been developed, which are easier to perform and have higher accuracy parameters [20]. It was also demonstrated that specific antibody serum levels present a direct correlation with clinical severity [21] and decrease with an efficacious antifungal treatment [22], which correlates with the recovery of specific-cell-mediated immunity, characterized by increased IL-2 and decreased IL-10 lymphocyte secretion [23]. These findings provided support for the use of specific anti-*Paracoccidioides* spp. serum antibody levels in the follow-up of patients under treatment as an important criterion of cure. These findings highlight the importance of serological tests in diagnosing, characterizing the severity of, and curing PCM patients. In addition, the DID test was defined as the reference test for evaluating the specific antibody serum levels in these patients [24]. However, this test has a relatively low degree of sensitivity, detecting antibodies only with a minimum concentration of 3 µg/mL [25], and has been shown to be less effective in PCM diagnosis in some specific clinical situations, like immunosuppressed patients [6], PCM with the involvement of some internal organs such as the CNS [26], and PCM relapse [10]. These findings led us to evaluate the DB assay and qPCR at the different PCM stages.

Our results showed a low sensitivity of the qPCR in the diagnosis of confirmed PCM patients and similar results between the DB and DID tests in the diagnosis and follow-up of these patients. However, the DB assay detected a lower concentration of specific serum antibodies and showed higher sensitivity than the DID test in the diagnosis of relapses.

The antigen used was a filtrate culture obtained from the *P. brasiliensis* B.339 strain, a strain that belongs to the *P. brasiliensis* complex, classified as S1 [27] or PS3 [28], which is reached in 43 kDa glycoprotein (gp43). The anti-*P. brasiliensis* IgG reacts with four major antigenic components of 70, 52, 43, and 20–21 kDa. The gp-43 marker is the predominant IgG reactive antigen, which is recognized by 100% of the patients’ sera, while gp70 is recognized by 96% [29,30]; both are PCM markers. In addition, IgG was identified by the DB test at a higher frequency than IgM and IgA in the PCM patients at the time of admission [31]. Thus, the choice of this antigen–culture filtrate from the *P. brasiliensis* B.339 strain and of anti-IgG antibodies for DB standardization is well justified.

The choice of the antigen plays a central role in a serological test. The molecular identification of clinical isolates of *Paracoccidioides* spp. revealed a high incidence of *P. lutzii* in central–western Brazil, suggesting that this finding could explain the presence of many confirmed PCM cases by identifying the fungus in clinical specimens, but with negative serology for the DID test using antigens from *P. brasiliensis* complex strains. This hypothesis was confirmed by Gegembauer et al. (2014) [30], who demonstrated the need to use species-specific antigens for PCM diagnosis caused by *P. lutzii*. However, little is known about the serological reactions of different species within the *P. brasiliensis* complex. A recent study revealed that sera from PCM patients due to *Paracoccidioides americana* infection reacted to antigens extracted from the Pb B.339 strain. Nonetheless, further studies on the other species within the *P. brasiliensis* complex are still necessary.

The semi-quantitative DID test is commonly used in the routine diagnosis of PCM in most clinical laboratories. This test showed a sensitivity of 90%, a specificity of 100%, a positive predictive value of 100%, a negative predictive value of 85.1%, and an accuracy of 93.6% in a study conducted on serum samples from 401 PCM patients from Botucatu upon admission (pre-treatment) [19]. It proved to be a simple, easy-to-perform, and cost-effective method. Although its results are available after four days, which is much less than for the culture of clinical specimens, this time can be too late for many cases. The relevant characteristics are its high specificity, positive predictive value, and usefulness in treatment monitoring. This serological test was used as a reference control in the present study.

The DB assay was found to be easy to perform, and its standard operational protocol allows for its inclusion in routine clinical laboratories. The results can be read without needing special equipment, unlike ELISA or immunoblotting [10], and are ready in only five hours, contrasting with the four days necessary for the DID test.

The accuracy parameters of the DB assay and DID test are very similar, with specificity, the positive predictive value, and the positive and negative likelihood ratios as the main characteristics. However, the sensitivity of both the DB (78.6%) and DID (73.8%) tests was lower than that shown in a previous study (90.0%) [19] carried out in the same region. These divergent findings could be explained by the number of patients evaluated (42 in the present study and 401 in the previous study) and by the number of cases with the mild form, which was proportionally higher in the present study (14%). As antibody serum levels are directly correlated with severity [21] and the antibodies of the IgG class are the last to be produced [31], mild cases usually present low titers or can be non-reagent.

Although the positive predictive value (PPV) of the DID and DB tests was 100%, this finding was observed in a sample with a PCM prevalence of 50%, which is much higher than that observed in populations in endemic areas. However, the high positive likelihood ratio plays a fundamental role in the interpretation of these tests because, differently from PPV, it is not influenced by disease prevalence [32].

Studies on semi-quantitative DB assays for the diagnosis of PCM are scarce. The semi-quantitative dot immunobinding test using a purified gp43 antigen showed a sensitivity of 100%, which is higher than our results with culture filtrate antigen [13]. The evaluation of the DB assay in the 27 confirmed PCM patients showed the identification of the immunoglobulins IgG in 81.5%, IgA in 51.9%, and IgM in 51.8% of the cases. Treatment with itraconazole led to a decrease in these frequencies [31].

A qualitative DB assay was also evaluated in a small number of patients using the recombinant protein p27 from *P. brasiliensis*, which is easier and less expensive to produce, and all the sera were reagents [33]. The qualitative DB assay for PCM diagnosis demonstrated higher sensitivity (91% vs. 73%) and a higher negative predictive value (90% vs. 76%) than the DID test [14]. Recombinant antigens (recombinant gp43 [34] and gp43∆Nt [35]) were also used in the DB assay to diagnose PCM patients with promising results.

Some studies suggested that the lack of reactivity of PCM patients’ sera in DID tests could be explained by low antibody concentrations below the method’s detection capacity [36], variations in the expression and isoforms of gp43 in different *P. brasiliensis* strains [37], and the low avidity of the IgG2 class of antibodies [38]. DID tests have low sensitivity, detecting only serum levels of at least 3 µg/mL [25]. In the present study, the absence of reactivity, both in the DID test and DB assay, was observed in the clinical cases considered mild. The IgG levels of these patients were probably still reduced, but below the detection limits. The DID test was positive for the presence of IgM, IgA, IgE, and IgG, with IgG being the last to be produced but persisting over a longer period and at higher titers [31]. We used anti-IgG goat immunoglobulin labeled with peroxidase to identify the presence of antibodies, which could explain why a delayed production of IgG or its low concentration could lead to the absence of reactivity in these sera. On the other hand, it was demonstrated that the patients with the most disseminated and severe forms had the highest serum concentrations of IgG [21], which could explain the absence of reactivity in these sera from the patients with mild disease. The presence of PCM patients infected with *P. lutzii* could also explain the low DID and DB sensitivity, which could not be analyzed because the fungi were not isolated from the patients’ clinical samples. However, only fungi of the *P. brasiliensis* complex were identified in a previous study performed on isolates from armadillos captured in this region and from clinical specimens of other patients admitted to our hospital, making this possibility unlikely [39].

We found no cross-reactivity in the sera of the patients with histoplasmosis, aspergillosis, cryptococcosis, and tuberculosis when evaluating the semi-quantitative DB assay in the serum of the patients with other systemic mycoses or tuberculosis using the antigen dilution standardized by us. Taborda et al. (1994) [13], in a semi-quantitative DB assay using purified gp 43 as the antigen, showed cross-reactivity in the sera of patients with Jorge Lobo’s disease, which was not observed with deglycosylated gp 43.

The qPCR conducted on the blood samples showed much lower sensitivity than the serological DID test and DB assay. This finding is in line with Buitrago et al. (2009) [15], who also observed low positivity in the blood samples and negativity in all the serum samples analyzed. The authors found positive results in sputum and fragments from affected tissue. In addition, positive clinical specimens turned negative after antifungal treatment. Using conventional PCR, San-Blas et al. (2005) [40] and Dias et al. (2012) [41] also found positivity in tissue fragments. The results of the present study suggest that *P. brasiliensis* fungemia is of short duration, making it challenging to identify its DNA in the blood. However, *Paracoccidioides* antigens were detected in biological fluids—urine [42] and serum [26]—demonstrating its circulation.

We found that the antibody serum levels decreased after introducing the antifungal treatment, both in the DID and DB tests. However, we detected a difference regarding the clinical form. The AF patients showed a regression of DID titers to non-reagents and DB titers to non-reagents or undiluted reagents in the studied period, differing from CF patients, whose DB titers persisted as reagents, while their DID titers decreased to non-reagents. Thus, a larger number of patients should be evaluated to define the antibody serum level determined by DB tests, which could be a reference for treatment discontinuation.

Cured PCM patients who present clinical manifestations compatible with or suggestive of a relapse constitute a clinical challenge. Furthermore, evaluating possible differential diagnoses caused by another infectious or non-infectious disease is mandatory. However, in the PCM patients with a relapse confirmed by mycological examination, the serological tests had low positivity, only 45% according to the DID test and 65% according to the ELISA, which are much lower than those shown by the same patients at the time of admission [10]. The semi-quantitative DB assay performed in 10 other relapsed patients had a 90% positivity during relapse, which was 30% in the DID test. These findings demonstrate that a DB assay is the serological test of choice for diagnosing relapses in PCM. However, it is important to reinforce that other diseases should be investigated to exclude comorbidity, such as cancer of solid organs [43,44] and tuberculosis [45].

In conclusion, our study with PCM patients demonstrated that a semi-quantitative DB assay is easy to perform and generates results in five hours, with an accuracy similar to the DID test in the diagnosis at the time of admission, no cross-reactions with several infectious diseases, and, finally, higher sensitivity for diagnosing relapses.

This study has limitations in the number of cases evaluated in the diagnosis and follow-up of the PCM patients and in the incidence of cross-reactions with other diseases according to the DB assay, which is ongoing. The future direction is evaluating DB accuracy in the cerebrospinal fluid of non-immunosuppressed patients and in the serum of patients with comorbidities that lead to disturbances of cell-mediated immunity, such as AIDS and neoplastic diseases.

## Figures and Tables

**Figure 1 microorganisms-12-00351-f001:**
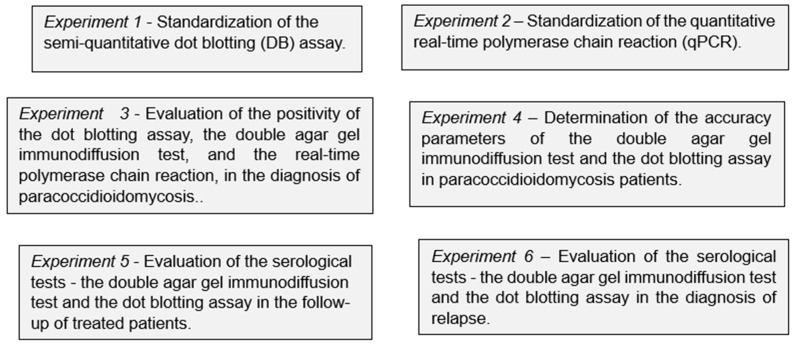
Infographic presentation of the six independent experiments carried out in the evaluation of the dot blotting assay and the quantitative real-time polymerase chain reaction in paracoccidioidomycosis patients compared to the double agar gel immunodiffusion test.

**Figure 2 microorganisms-12-00351-f002:**
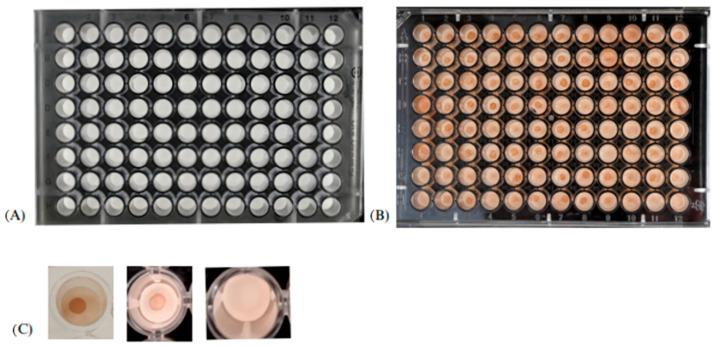
Illustrative image of the different phases of the dot blotting assay. (**A**) Pre-test plate; (**B**) plate after the addition of the DAB chromogen; (**C**) reading—the presence of strong and weak circles indicates a positive reaction, while the absence of a circle indicates a negative reaction.

**Figure 3 microorganisms-12-00351-f003:**
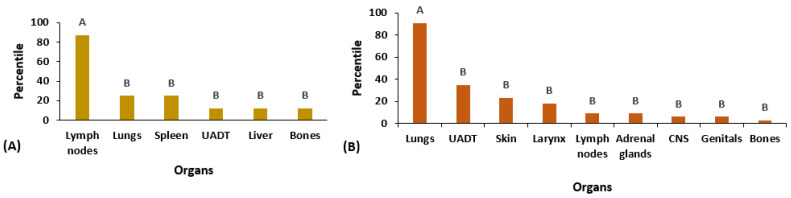
Distribution of eight patients with the acute/subacute (AF) form of paracoccidioidomycosis and 34 cases with the chronic form (CF) according to the organs involved at admission (pre-treatment). (**A**) Patients with the AF. (**B**) Patients with the CF. Frequencies followed by the same letter do not differ (*p* > 0.05), while different letters indicate statistically significant differences (*p* ≤ 0.05).

**Figure 4 microorganisms-12-00351-f004:**
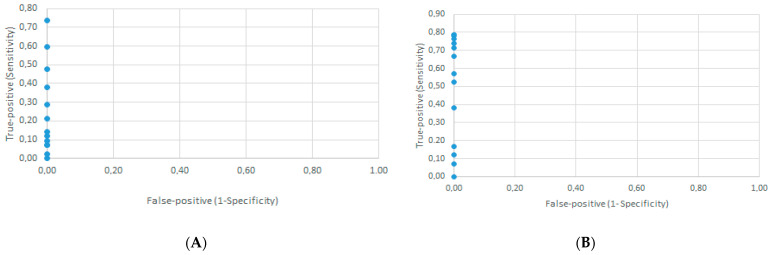
Determination of the cut-off point by the receiver operator characteristic curve using the results from 42 confirmed paracoccidioidomycosis patients at admission and 42 healthy individuals (controls). (**A**) Semi-quantitative serological double agar gel immunodiffusion test, false-positive rate against true-positive rate for 13 possible cut-off points. (**B**) Semi-quantitative serological dot blotting assay, false-positive rate against true-positive rate for 18 possible cut-off points.

**Figure 5 microorganisms-12-00351-f005:**
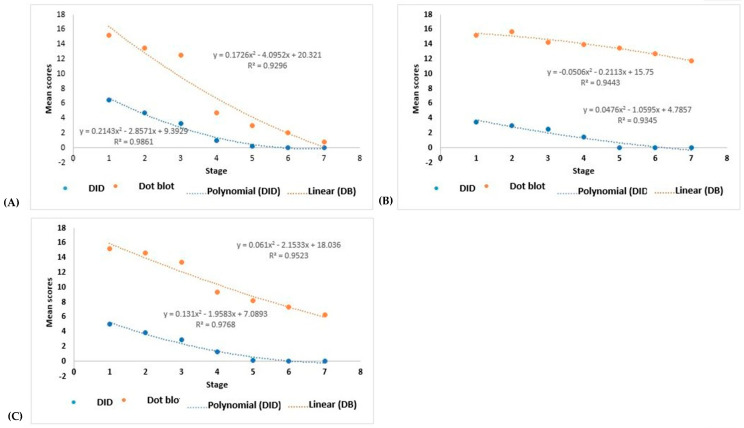
Polynomial regression representing the variation in scores of serum antibody titers against *Paracoccidioides brasiliensis*, determined by the dot blotting assay and the double agar gel immunodiffusion test, in response to antifungal treatment. Evaluation at different stages: 1—pre-treatment; 2—one month of treatment; 3—2 to 3 months of treatment; 4—4 to 10 months of treatment; 5—11 and 12 months of treatment; 6—14 and 15 months of treatment; 7—treatment >15 months. (**A**) Evaluation of four patients with the acute/subacute form; (**B**) evaluation of four patients with the chronic form; (**C**) evaluation of all eight patients.

**Table 1 microorganisms-12-00351-t001:** Dot blotting assay and double agar gel immunodiffusion test. Conversion of the serum antibody levels, determined as dilutions, into scores.

Titer	Score	Titer	Score
Non-reagent	0	512	10
1 (reagent undiluted)	1	1024	11
2	2	2048	12
4	3	4096	13
8	4	8192	14
16	5	16,384	15
32	6	32,768	16
64	7	65,536	17
128	8	131,072	18
256	9	262,144	19

**Table 2 microorganisms-12-00351-t002:** Characterization of 42 patients with active paracoccidioidomycosis according to sex, age, clinical form, and severity, included in the evaluation of diagnostic methods.

Clinical Form (Number of Cases)	Severity (Number of Cases)	Sex (Number of Cases)	Age [Mean (Years)]
Acute/subacute (8)	Moderate (2)	Male (2)	16.5
	Severe (6)	Female (3) + Male (3)	22
Chronic (34)	Mild (6)	Female (3) + Male (3)	53.0
	Moderate (24)	Female (3) + Male (21)	56.0
	Severe (4)	Male (4)	61.0

**Table 3 microorganisms-12-00351-t003:** Distribution of 42 paracoccidioidomycosis patients with active disease at admission (pre-treatment), 8 of them with the acute/subacute form and 34 with the chronic form, according to specific serum antibody levels determined by the double agar gel immunodiffusion test (DID) and the dot blotting assay (DB), as well as the severity. Data are presented as the median and the 1st and 3rd quartiles of the scores. A Mann-Whitney test and Kruscal-Wallis test were used.

Clinical Form	Severity	DID		DB	
		median	quartile	median	quartile
Acute form	Moderate	1.5	[1.0;2.0]	13.0	[12.0;14.0]
	Severe	2.0	[0.0;9.0]	15.5	[12.0;18.0]
*p* value		0.86		0.43	
Chronic form	Mild	0.0 B	[0.0;0.0]	0.0 B	[0.0;0.0]
	Moderate	3.0 A	[1.0;4.0]	14.0 A	[11.0;15.0]
	Severe	5.0 A	[2.5;6.5]	14.0 AB	[7.0;15.0]
*p* value		0.005		0.008	

Uppercase letters compare medians within the same column; medians followed by the same letter do not differ from each other (*p* > 0.05), while those followed by different letters present statistically significant differences (*p* ≤ 0.05).

**Table 4 microorganisms-12-00351-t004:** Results of the dot blotting assay (DB) and the double agar gel immunodiffusion test (DID) performed in the sera of 42 patients with active paracoccidioidomycosis (PCM) at admission (pre-treatment) and in the sera of 42 healthy blood donors (controls) from the same region where the patients came from.

	DB			DID		
Test result	PCM patients	Controls	Total	PCM patients	Controls	Total
Reagent	33	-	33	31	-	11
Non-reagent	09	42	51	11	42	53
Total	42	42	84	42	42	84

**Table 5 microorganisms-12-00351-t005:** Accuracy parameters of the dot blotting assay (DB) and the double agar gel immunodiffusion test (DID) calculated using the results obtained from 42 patients with confirmed paracoccidioidomycosis at admission (pre-treatment) and 42 controls—donors from the Blood Bank of Botucatu.

Test	Sensitivity	Specificity	PPV	NPV	Accuracy	PLR	NLR
Dot blotting	78.6%	100%	100%	82.35%	89.28%	330	0.21
DID	73.8%	100%	100%	79.24%	86.9%	310	0.26

PPV—positive predictive value; NPV—negative predictive value; DID—double immunodiffusion in agar gel. PLR—positive likelihood ratio; NLR—negative likelihood ratio.

**Table 6 microorganisms-12-00351-t006:** Characterization of eight paracoccidioidomycosis patients undergoing antifungal treatment according to clinical form, severity, gender, and age.

Clinical Form (Number of Cases)	Severity (Number of Cases)	Sex (Number of Cases)	Age (Mean (Years))
Acute/subacute (4)	Moderate (2)	Male (1) + Female (1)	10.0
	Severe (2)	Female (2)	15.5
Chronic (4)	Mild (2)	Male (2)	58.0
	Moderate (2)	Male (2)	53.0

**Table 7 microorganisms-12-00351-t007:** Characterization of 10 patients with relapse of paracoccidioidomycosis according to gender, age, clinical form, and severity, determined at the first admission.

Clinical Form (Number of Cases)	Severity (Number of Cases)	Sex (Number of Cases)	Age (Mean (Years))
Acute/subacute (1)	Severe (1)	Female (1)	85.0
Chronic (9)	Mild (1)	Male (1)	60.0
	Moderate (7)	Male (7)	46.0
	Severe (1)	Male (1)	35.0

**Table 8 microorganisms-12-00351-t008:** Comparative evaluation of dot blotting assay and double agar gel immunodiffusion test in paracoccidioidomycosis patients with relapse. Evaluation in relation to the test and the stages of the disease. McNemar’s Test.

DID vs. DB	DID + DB+	DID-DB-	DID + DB-	DID-DB+	DID+ (%)	DB+ (%)	*p* Value
Pre-treatment	8	-	-	2	80	100	0.48
Relapse	3	1	-	6	30	90	0.04

Pre-T (1) vs. relapse (2)	1 + 2+	1-2-	1 + 2-	1-2+	Pre-treatment positive (%)	Relapse positive (%)	
DID	3	2	5	-	80	30	0.07
DB	9	-	1	-	90	90	1.00

Pre-T—pre-treatment; DID—double agar gel immunodiffusion test; DB—dot blotting.

## Data Availability

The data presented in this study are available on request from the corresponding author. The data are not publicly available due to [Individual data belongs to the patients and are unavailable due to ethics restrictions].

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
