# Peer review of "Standardization of Semi-Quantitative Dot Blotting Assay—Application in the Diagnosis, Follow-Up, and Relapse of Paracoccidioidomycosis"

_microorganisms, 2024, doi:10.3390/microorganisms12020351_

Round 1

Reviewer 1 Report

Comments and Suggestions for Authors

Were patients with others infections diseases, on treatment or not? This information is important because the antigent can be high or low according to the course of treatment. 

Author Response

No, the patients with comorbidities wer excluded of the study, as you can see in the exclusion criteria.

Reviewer 2 Report

Comments and Suggestions for Authors

Reviewer Report:

This paper aims to standardize the semi-quantitative dot blotting assay (DB) and to identify the real-time polymerase chain reaction (qPCR) to detect specific antibodies for Paracoccidioides brasiliensis and its DNA in PCM patients.

This study was conducted in 60 PCM patients; 42 were treatment-näive, eight were under antifungal treatment follow-up (four with AF and four with CF), and 10 were at the moment of a relapse so authors evaluated the serological progress during treatment in 8 patients and used 10 patients for relapse diagnosis in using the Pb B.339 strain antigen.

The manuscript is written comprehensively enough to be understandable despite of the subject complexity; the authors addressed this aim by showcasing the workflow of the designed experiments used in this study and showing in figure 1.

The paper stated the purpose, discussion and global implication are clearly stated and consistent with the rest of the manuscript; authors provided the required tests, analysis and enough information in their discussion by using a good number of important articles talked about the subject.

The authors addressed their hypothesis and opinion in a reproducible way and they proved their results through all the required experiments and calculation. The results were presented in a clear way which facilitate in reaching a conclusion elucidates positive results found in sputum and fragments from affected tissue. In addition, positive clinical specimens turned negative after antifungal treatment. They also found that antibody serum levels decreased after introducing the antifungal treatment, both with DID and DB., however with a further study, a higher number of patients should be evaluated to define the antibody serum level determined by DB, which could be the reference for treatment discontinuation.

The abbreviations should be explained at the first place they are mentioned.

In vitro, in vivo, et al.: should be written in italic.

No plagiarism has been detected.

References: The authors should follow the journal guidelines for some references.

Year: should be in Bold, Volume: should be in italic

Comments on the Quality of English Language

Minor editing of English language required

Author Response

The introduction and the references cited seems to be satisfactory to introduce the study. I know that there is a very rich literature about serological diagnosis of paracoccidioidomycosis, which I avoid to be objective. As the other two reviewers also considered satisfactory these points, I would like to suggest that you agree with the text presented.

The references were reviewed and corrected in the text. The text, written in English, was purified/corrected by EscritaLab (escrita.cientifica.lab@gmail.com).

Reviewer 3 Report

Comments and Suggestions for Authors

Summary:

The authors have standardized a semi-quantitative dot blotting (DB) assay for diagnosis of paracoccidioidomycosis (PCM) and evaluated its performance compared to double immunodiffusion (DID) test and PCR in 42 patients at admission, 8 undergoing treatment follow-up, and 10 relapse cases. Overoll, the authors have conducted a well-designed study showing promising results for DB in PCM diagnosis and relapse detection. Addressing the following suggestions would strengthen the conclusions and clinical applicability of the DB assay.

Suggestions:

  1. Increase the number of relapse cases tested to at least 30-50 patients, if possible, to better substantiate the higher sensitivity of dot blotting (DB) over double immunodiffusion (DID) for detecting relapse.
  2. Expand the range of diseases tested for cross-reactivity to include other endemic mycoses (blastomycosis, coccidioidomycosis, histoplasmosis), autoimmune diseases (rheumatoid arthritis, lupus), and chronic infections (TB, leprosy) to further validate specificity of the DB assay.
  3. Conduct a head-to-head comparison of DB versus DID for diagnosis in 10-20 immunosuppressed patients, especially those with HIV/AIDS, to determine if DB has higher sensitivity in this population.
  4. Follow up an additional 10-20 patients on treatment for at least 18-24 months with serial DB and DID testing to better delineate long-term antibody kinetics and clarify when DB titers become negative.
  5. Analyze DB titer declines at 6, 12, 18 and 24 months to propose a threshold drop (e.g. 4-fold reduction) that could indicate treatment response and guide duration of monitoring.                                            I

Author Response

  1. Question

Answer. The cases of relapse have been observed five years after the discontinuation of the antifungal treatment, usually in 5% of the patients, independently of the clinical form. The increase of this sample will be performed along the time.

  1. The increase of the samples evaluated was proposed at the end of the article, including to evaluate the incidence of cross reactions. Regarding the diseases cited, blastomyosis is not available in Brazil, coccidioidomycosis has been reported only in the Northeast Brazil and histoplasmosis is much more frequent in AIDS-patients in our region – the incidence in non-immunosupressed patients is low. Thus, this data should be reviewed in the next years or in other laboratory that standardize this test.
  2. It is an interesting suggestion. However, the prevalence of paracoccidioidomycosis in immunosuppressed patients is very low, including in AIDS-patients. I would like to take into consideration your suggestion for another study, after collecting this kind of patients.
  3. This suggestion was written at the end of the manuscript, because it will demand at least two more years of study.
  4. The definition of a serological threshold drop depends on the increase of the number of patients with follow-up for long time, comparing patients with and without relapse. This study should be developed as another project, and it is in our line of research. The definition of a biomarker for discontinuation of the therapy continues to be a challenge in paracoccidioidomycosis and one of our targets.

Round 2

Reviewer 3 Report

Comments and Suggestions for Authors

The authors did not sufficiently address my concerns; however, I find the explanation to be acceptable. It would be beneficial to witness a follow-up study in the future.

Author Response

The suggestions of the reviewers were accepted - the use of italics was corrected; the increase of the samples to analyse the cross reactions was also included in the text, line 556